# ABHD4-Regulating RNA Panel: Novel Biomarkers in Acute Coronary Syndrome Diagnosis

**DOI:** 10.3390/cells10061512

**Published:** 2021-06-16

**Authors:** Sara H. A. Agwa, Sherif Samir Elzahwy, Mahmoud Shawky El Meteini, Hesham Elghazaly, Maha Saad, Aya M. Abd Elsamee, Rania Shamekh, Marwa Matboli

**Affiliations:** 1Clinical Pathology and Molecular Genomics Unit, Medical Ain Shams Research Institute (MASRI), Faculty of Medicine, Ain Shams University, Cairo 11382, Egypt; 2Cardiovascular Medicine Department, Faculty of Medicine, Ain Shams University, Cairo 11382, Egypt; Dr.sherifelzahwy@med.asu.edu.eg; 3Department of General Surgery, The School of Medicine, University of Ain Shams, Cairo 11382, Egypt; mahmoud_elmeteini@med.asu.edu.eg; 4Oncology Department, Medical Ain Shams Research Institute (MASRI), Faculty of Medicine, Ain Shams University, Cairo 11382, Egypt; heshamelghazaly@med.asu.edu.eg; 5Biochemistry Department, Faculty of Medicine, Modern University for Technology and Information, Cairo 11382, Egypt; maha.saad@medicine.mti.edu.eg; 6Biochemistry and Molecular Genomics Unit, Medical Ain Shams Research Institute (MASRI), Faculty of Medicine, Ain Shams University, Cairo 11382, Egypt; aya_ana2025@yahoo.com; 7Department of Pathology, University of South Florida, Tampa, FL 33620, USA; Rania.shamekh@gmail.com; 8Medicinal Biochemistry and Molecular Biology Department, Faculty of Medicine, Ain Shams University, Cairo 11382, Egypt

**Keywords:** acute coronary syndrome, RNA, diagnosis, serum, troponin, bioinformatics, glycerophospholipid

## Abstract

Background: Acute coronary syndrome (ACS) is a major cause of death all over the world. STEMI represents a type of myocardial infarction with acute ST elevation. We aimed to assess the predictive power of potential RNA panel expression in acute coronary syndrome. Method: We used in silico data analysis to retrieve RNAs related to glycerophospholipid metabolism dysregulation and specific to ACS that results in the selection of Alpha/Beta hydrolase fold domain4 (*ABHD4*) mRNA and its epigenetic regulators (Foxf1 adjacent noncoding developmental regulatory RNA (*FENDRR) lncRNA, miRNA-221, and miRNA-197*). We assessed the expression of the serum RNA panel in 68 patients with ACS, 21 patients with chest pain due to non-cardiac causes, and 21 healthy volunteers by quantitative real-time polymerase chain reaction. Results: The study data showed significant down regulation in the expression of the serum levels of *FENDRR* lncRNA and miRNA-221-3p by 120-fold and 22-fold in Unstable angina (UA) in comparison with healthy volunteers, and by 8.6-fold and 2-fold in ST segment elevation myocardial infarction (STEMI) patients versus UA; concomitant upregulation in the expression of *ABHD4* mRNA and miRNA-197-5p by 444-fold and 10-fold in UA compared with healthy volunteers, and by 1.54-fold and 4.5-fold in STEMI versus unstable angina. Performance characteristics analysis showed that the *ABHD4*-regulating RNA panel were potential biomarkers for prediction of ACS. Moreover, there was a significant association between the 2 miRNAs and *ABHD4* mRNA and the regulating *FENDRR* lncRNA. Conclusion: Collectively, *ABHD4* mRNA regulating RNA panel based on putative interactions seems to be novel non-invasive biomarkers that could detect ACS early and stratify severity of the condition that could improve health outcome.

## 1. Introduction

Acute coronary syndrome is a prime reason of hospital admission and death worldwide [1]. Myocardial infarction definition necessitates necrosis of the cardiac myocytes with abnormal levels of plasma cardiac troponin [2,3]. Early diagnosis of ACS allows early reperfusion therapy, resulting in decrease in the death rate [4]. Diagnostic biomarkers of ACS have widely emerged through the study of known myocardial proteins [5]. Currently, many biomarkers as cardiac myoglobin, creatine kinase-MB, and troponins are widely used in the clinical diagnosis with unfortunately many false elevations due to skeletal muscle injury. From these biomarkers, troponins are now considered as the corner stone in ACS diagnosis but it has the following limitations: (1) it begins to increase within 3–4 h after the incidence of myocardial ischemia, limiting ACS diagnosis within first 1–2 h, (2) it is elevated in the chronic renal failure patients and has limited use in myocardial re-infarction due to its long half time [6,7]. Yet, the need to find novel biomarkers with high accuracy is always present, to allow earlier diagnosis with decreased mortality and complications [8,9]. Dysregulation of lipid metabolism leads to several chronic diseases especially cardiovascular disease. Shedding the light on the enzymatic mechanisms controlling lipid metabolism is crucial for successful innovative diagnostics and drug discovery in human diseases. Alpha/Beta hydrolase fold domain containing (*ABHD*) proteins family have common roles in lipid metabolism [10]. They include proteases, esterases, lipases, dehalogenases, and epoxide hydrolases [11]. ABHD protein family members have the ability to metabolize different glycerophospholipid types that act as key intermediates in cellular signaling and neurotransmission [12]. Recently, ABHD protein family mutations have been linked to inborn errors of lipid metabolism with subsequent complications. ABHD protein family is considered as a predictor of plasma phospholipid levels in humans [13].

Plasma glycerophospholipid and lysophosphatidic acids (LysoPAs) are crucial lipid mediators, and the main risk factors for cardiovascular diseases including ACS [14]. LysoPA stimulates the adhesion of molecules and chemokines expression in the endothelial cells, the smooth muscle cells migration, and platelets activation [15]. Several studies highlighted the activations of phospholipase A2 and phospholipase D15 in myocardial ischemia [16]. Alpha/Beta Hydrolase Domain-Containing Protein 4 (*ABHD4*) gene is a main regulator of phospholipid metabolism in mammals having hydrolase and lysophospholipase activity [17]. *ABHD4* deacylates N-acylphosphatidylethanolamines and N-acylphosphatidylserines to hydrolyze saturated and unsaturated N-acyl chains. *ABHD4* (–/–) mice show decrease in brain glycerophosphoethanolamine and lysophosphatidylserines [18].

miRNAs are known to regulate important complex gene regulatory pathways related to the development of cardiovascular system [19].Certain patterns of miRNA expression plays a prime role in myocardial infarction, in addition to the fact that many cardiac miRNAs are dysregulated in patients with ACS [20,21]. Several miRNAs have been studied in ACS, such as *miRNA-208a, miRNA-126, miRNA-122-5p, miRNA-19a,* and *miRNA-1*, which have been recognized as new biomarkers in early ACS diagnosis [22]. So, further studies of the relation between target genes and miRNA are needed for better comprehension of MI pathology and potential biomarker discovery. Additionally, dysregulation of lncRNA expression is involved in many diseases including cancers [23] and cardiovascular diseases [24]. For example, *Mirt1 lncRNA* [25], *lncRNA KCNQ1OT1* [26], and *aHIF lncRNA* and *ANRIL lncRNA* [27] have been related to myocardial infarction.

In this study, we used in silico data analysis to explore a new mechanistic signaling pathway based on putative interactions between RNAs in ACS. We chose *ABHD4*-regulating RNA panel related to glycerophospholipid metabolism and related to ACS that could be a potential biomarker in early ACS diagnosis and detection of myocardial ischemia in unstable angina with low troponin level or in STEMI patients with high troponin level. We measured the expression of the serum RNA panel in ACS patients group, patients presented with chest pain due to non-cardiac causes, and healthy volunteers.

We have selected *ABHD4* mRNA as a crucial player in glycerophospholipid metabolism gene highly correlated with cardiovascular complications, especially ACS based on two approaches. Firstly, bioinformatics analysis was used to confirm the expression of *ABHD4* mRNA in ACS based on novelty, gene ontology enrichment that confirm its correlation with lipid metabolism regulation, and the basal expression in heart. Taken together, the public microarray gene expression databases confirmed such selection. The second approach was a literature review [1,2,3,4] of the limited data available about the role of *ABHD* family as predictor of glycerophospholipid in humans, which has a strong correlation with ACS. Both bioinformatics analysis and literature review suggested a possible role of *ABHD4* mRNA in ACS. Afterwards, we retrieved data about miRNA regulation of *ABHD4* mRNA and identified (*miRNA-197-5p and miRNA-221-3p*) based on both in silico data and literature review [28,29,30,31,32,33] that affirm that the chosen miRNAs were linked to lipid metabolism regulation as previously confirmed in ACS, targeting *ABHD4* mRNA. Lastly, we aim to construct an integrated and genetically linked mRNA–miRNA–lncRNA panel to increase the chance of usefulness of the chosen panel in ACS management and novel implications for targeting ABHD enzymes in the treatment or prevention of lipid-metabolism-related disease, especially ACS. We have selected *FENDRR* lncRNA based on both in silico data that confirm complementarity binding between the selected RNA panel members and literature search that highlighted its role in inflammation, fibrosis, and cardiovascular diseases [34,35].

## 2. Materials and Methods

### 2.1. Study Population

This study is approved by the Ain Shams ethical committee, faculty of medicine. All the patients were recruited from the cardiovascular (CVS) department, Ain Shams University in the period from November 2017 till October 2018. The study includes 68 acute coronary syndrome patients including UA (*n* = 21), STEMI (*n* = 31) patients, and NSTEMI (*n* = 16); 21 patients with non-cardiac chest pain based on the output of coronary angiography; and 21 healthy volunteers with normal ECG and no history of CVS disease seeking routine health checkups with matched sex and age to the patients’ groups. Informed consent was taken from all participants.

ACS was diagnosed through assessing cardiac troponin levels, creatine kinase-MB (CK-MB) together with clinical symptoms consistent with ACS within 6 h of chest pain and underwent primary Percutaneous Coronary Intervention (PCI). ACS was diagnosed on the basis of ischemic symptoms, a pathological Q wave, and an increased cTnI (cardiac troponin I) with CK-MB expression according to American College of Cardiology/American Heart Association (2018 ESC/ACC/AHA/WHF Fourth Universal Definition) guidelines. Subjects with end-stage renal failure, liver disease, cardiomyopathy, hemorrhagic disorders, immunological diseases, chemotherapy or radiotherapy, or inflammatory bowel disease, chronic myopathy and cancer were excluded from the study.

Blood samples were collected in the first 6 h of chest pain onset. Continuous assessment of CK-MB and hs-cTnT were done. Samples were centrifuged at 4000 rpm for 20 min, and the sera samples were kept in aliquots and stored at −70 °C into DNase-/RNase-free eppendorf tubes.

### 2.2. In Silico Data Analysis

We have chosen *ABHD4* gene as it is linked to glycerophospholipid metabolism and ACS from GeneCards^®^: The Human Gene Database and Human Protein Atlas database based on novelty and basal expression in normal heart (Appendix A). Then, based on high complementarity binding site numbers, relation to lipid metabolism regulation, and relation to coronary syndrome and expression in heart, *miRNA-221-3p* and *miRNA-197-5p* miRNAs were selected through miRWalk database. Pathway enrichment analysis of both miRNAs ensured their relation to lipid metabolism, apoptosis, cytokines, and inflammation that are closely linked to ACS (Appendix A). Finally, *FENDRR lncRNA* was selected as a master control player of the previously selected genes through non-code database. Sequence alignment was done between the *FENDRR lncRNA* and the selected miRNAs to confirm the in silico prediction (Appendix A).

### 2.3. Purification of Total RNA Including miRNAs from Sera Samples and Quantitaive Real Time PCR (RT-qPCR)

Total RNA was purified from the sera samples by miRNEasy extraction kit (Qiagen, Hilden, Germany) according to the kit manual. Concentration & purity of RNA was analyzed by NanoDrop (Thermo Scientific, Waltham, MA, USA) and with Invitrogen™ Qubit™ 3.0 Fluorometer (Termo Fisher, Waltham, MA, USA). Equal amounts of RNA were used for reverse transcription (RT) using the TaqMan miRNA Reverse Transcription Kit and for amplification by qPCR, using TaqMan MicroRNA Assays of the selected miRNAs, ABHD4 Taqman probe and TaqMan universal mastermix (Applied Biosystems, Foster City, CA; Termo Fisher, Waltham, MA, USA), U6 sn RNA, and endogenous control.

FENDRR lncRNA in the sera samples were assessed using miScript II RT Kit (Qiagen, Hilden, Germany) to synthesize cDNA, followed by RT2 SYBR Green ROX qPCR Mastermix (Qiagen, Germany) and ACTB-1/beta actin (Hs-ACTB-1-RT2 QuantiTect Primer Assays, Qiagen, Germany) as endogenous control. Each sample was assessed in duplicate. Spike-in control cel-miRNA-39 (Qiagen, Germantown, MD, USA) was used for the normalization of miRNAs. Relative quantification of RNA panel expression was calculated by RQ = 2^−∆∆Ct^ using Livak method. RT-qPCR was done using Applied Biosystems 7500 FAST Real Time PCR system (Applied Biosystems, Foster City, CA, USA) thermal cycler with data analysis taking into consideration the negative expression if Ct value was more than 36 (details in Appendix A).

### 2.4. Statistical Analysis

Data was statistically analyzed using software package of statistical analysis version 25(SPSS25): median for non-parametric data, while mean ± SD for symmetrically distributed raw numerical data. One-way ANOVA, cross-tabulation chi-square test for number and percentage calculation, and Spearman correlation test were used as appropriate. The receiver operating characteristic (ROC) curve was used to evaluate the predictive value of the RNA panel in acute coronary syndrome.

## 3. Results

### 3.1. Biochemical and Clinical Markers in the Investigated Groups

Concerning the clinical and laboratory data (age, sex, smoking, hypertension, diabetes mellitus, and serum LDL, HDL, and total cholesterol), we did not find significant difference in the ACS, non-cardiac chest pain, and control groups (*p* > 0.05). On the contrary, we found highly significant difference in body mass index and serum levels of triglycerides, and creatinine among the three investigated groups (*p* < 0.05) (Appendix A).

### 3.2. Expression of the Serum RNAs Molecular Network

Serum levels of the chosen molecular network RNAs were assessed in samples of the 3 different groups to validate our retrieved in silico data. The expression was assessed through fold change (RQ) values. There was down regulation in the expression of *FENDRR lncRNA* and *miRNA-221* with concomitant upregulation in the level of *ABHD4* mRNA and *miRNA-197* in the ACS group compared with both non-cardiac chest pain group and healthy volunteers (*p* < 0.001) (Table 1, Figure 1A–D).

Using ROC curve analysis, we compared the ACS group to both chest pain group due to non-cardiac causes and healthy volunteer group. We found that the best cutoff values were 2.1 for *ABHD4* mRNA (AUC = 0.972), 2.55 for *FENDRR* lncRNA (AUC =0.949), 2.02 for *miRNA-221-3p* (AUC = 0.958), and 1.7 for *miRNA-197-5p* (AUC = 0.949). The measured sensitivities were 94.4%, 93.3%, 95.4%, and 89.3%, respectively. The aforementioned results point out that these optimal cutoff values could be used to discriminate between ACS from non-cardiac chest pain patients and healthy participants (Table 2, Figure 2A–F, Figure 3 and Figure 4).

Additionally, the expression pattern for ABHD4 mRNA regulating RNA panel in unstable angina and acute STEMI were compared with healthy participants. The serum levels of *FENDRR* lncRNA and *miRNA-221-3p* were increased by 120-fold and 22-fold in unstable angina compared with healthy participants, and by 8.6-fold and 2-fold in STEMI compared with unstable angina. Moreover, there was a concomitant upregulation in the expression of *ABHD4* mRNA and *miRNA-197-5p* by 444-fold and 10-fold in unstable angina compared with healthy participants, and by 1.54-fold and 4.5-fold in STEMI compared with UA, respectively. Additionally, at an optimal cut-off value of 13.55, 7.07, and 0.25 for *ABHD4* mRNA, *miRNA-197-5p*, and *FENDRR* lncRNA expression levels could potentially distinguish UA from patients with STEMI (AUC: 0.615, 0.773, and 0.774, respectively) (Figure 3) with sensitivity of 77.2%, 81%, and 71.4% and specificity of 52%, 70.2%, and 76.6%, respectively (Table 2, Figure 3A,B and Figure 4).

Contrary to the limited elevation of cardiac troponin levels in only 33% of UA patients, *ABHD4* mRNA regulating RNA panel were markedly elevated in 94%, 93%, and 89% for *ABHD4 mRNA, FENDRR lncRNA, and miRNA-197-5p*, respectively, of UA patients (Figure 4A,B). For acute STEMI patients, myocardial infarction was verified by persistent ST-segment changes and significant increase in cardiac troponin in all patients, similar to the marked differential expression of *ABHD4* mRNA regulating RNA in the STEMI patients compared with healthy controls (Figure 3 and Figure 4).

### 3.3. Correlation between Serum ABHD4 Regulating RNAs and Cardiac Troponin among the Study Groups

On the other hand, there was significant inverse correlation between *ABHD4* mRNA and both *FENDRR lncRNA and miRNA-221-3p* (*p* < 0.000). Furthermore, there was significant positive correlation between *ABHD4 mRNA and miRNA-197* (*p* < 0.000) (Table 3).

The downregulation of *FENDRR* lncRNA during ACS results in an increase in the expression of *miRNA-197-5p* and decrease in the level of *miRNA-221-3p*, which sequentially activates the *ABHD4* mRNA. Results also go in hand with the ontology bioinformatics evidence that (*FENDRR lncRNA, miRNA-197-5p, and miRNA-221-3p*) networks synergistically regulate the *ABHD4* mRNA expression in ACS, and thus shed light on a novel molecular mechanism in myocardial ischemia (Figure 4A,B and Figure 5).

## 4. Discussion

ACS is a main cause of disability and high death rate in developed countries [22], and ACS causes about 1/3 of all deaths in people older than 35 years in western countries [23]. The ECG is an important diagnostic tool in acute coronary syndrome; however, it lacks sensitivity and about 30–50% of patients may initially present symptoms with normal ECG [24]. Creatine kinase MB assessment is not only specific for MI and there are conditions with elevated CK-MB concentrations other than acute coronary syndrome. Several cardiac diseases as cardiac failure and arrhythmia may result in increased CK-MB concentration [25]. On the other hand, using cTn in ACS diagnosis requires careful inspection, as the diagnosis remains inconclusive for about 44% of patients who require additional modalities for diagnosis [26]. Thus, the ongoing search for novel early diagnostic biomarkers for ACS is unstoppable.

Recent studies found that ncRNAs including both miRNAs and lncRNAs are of value in the cardiovascular diseases diagnosis and treatment [27]. Target genes of certain miRNAs are often not validated. Moreover, several in silico predicted target genes may be physiologically irrelevant at lower physiological concentrations, or because of differences in localization between miRNAs and their targets, so we selected a RNA network including target gene, lncRNA, and miRNA to enrich our findings. We aim to evaluate the diagnostic accuracy of *ABHD4*-regulating RNAs (*miRNA-197-5p* as a marker of apoptosis; *miRNA-221-3p* as a marker of inflammatory response, cell adhesion and phospholipid metabolism dysregulation; and *FENDRR* lncRNA as a marker of myocardial development) for the early detection in patients with unstable angina with troponin level lower than the detection limit and patients who had more than fifty percent coronary artery occlusion. Results indicate that *ABHD4*-regulating RNAs were significantly dysregulated in UA and STEMI patients compared with healthy controls and that could detect symptomatic unstable angina patients with high sensitivity and specificity, and also estimate the severity of ACS with distinct differential expression in STEMI compared with unstable angina. *ABHD4*-regulating RNAs retrieval was based on a putative interaction and might be a novel biomarker in ACS.

*ABHD4* is a paralog of *ABHD5* [36], which is known to regulate lipolysis [37]; it is also known to interact with lipolysis regulatory proteins, to regulate autophagy and to have a role in energy and lipid metabolism [38]. The *ABHD* subfamily belongs to a large protein family that is characterized by the presence of α/β hydrolase fold [39]. *ABHD4* gene is a critical regulator of phospholipid metabolism in mammals [40]. ABHD4 plays a significant role in anoikis resistance [41]. In our study, we reported up regulation of serum *ABHD4* expression in ACS group compared with chest pain due to non-cardiac causes and healthy participants groups. To the best of our knowledge, it was first time to report association between *ABHD4* mRNA dysregulation and ACS.

Xue et al. found high levels of *miRNA-221-3p* in blood vessels with atherosclerotic patches [31]. In addition, Coskunpinar et al. reported that *miRNA-221-3p* expression was deregulated in AMI patients [32]. Jia et al. reported that lower concentration of circulating *miRNA-221* was significantly associated with coronary heart disease. In our study, we also reported downregulation of miRNA-221 expression in ACS patients.

*miRNA-197* is located on human chromosome 1, with high expression in platelets [18]. *miRNA-197* contributes to dyslipidemia associated with metabolic syndrome, resulting in coronary heart diseases [30]. Schulte et al. in a cohort study found that elevated levels of *miRNA-197* is predictive of cardiovascular death [41]. Additionally, *miRNA-197* was found to be correlated with myocardial fibrosis [32,41,42]. In this study, we reported the upregulation of *miRNA-197* expression in ACS group compared with chest pain due to non-cardiac causes and healthy volunteers groups.

*FENDRR* lncRNA (Foxf1 adjacent noncoding developmental regulatory RNA) is a crucial player in heart development [43]. Çekin et al. declared that FENDRR expression was lower by 7 folds in coronary artery plaques [44]. In our study, we reported downregulation of *FENDRR* lncRNA expression in ACS group compared with chest pain due to non-cardiac causes and healthy participants groups.

Study limitations include small sample size collected only from one medical facility in Egypt. At the in silico data analysis, it was found that both miRNAs could interact with the selected mRNA and lncRNA with a score > 0.95 at CDS binding site (Appendix A) that showed putative interaction with *ABHD4* but there are several mismatches. Mismatches are also visible in the sequence alignment between ABHD4 and lncRNA *FENDRR* (Appendix A). Thus, further in vitro and in vivo studies are still required to confirm our findings.

## 5. Conclusions

In spite of the availability of cardiac troponin, there is still an urgent demand for novel biomarkers with higher diagnostic accuracy for early detection of ACS patients with troponin levels lower than the detection limit. In summary, *ABHD4* regulating RNA panel based on putative interactions (marker of glycerophospholipd metabolism, cell damage, apoptosis, inflammation) and gene expression was assessed in sera samples from patients presented with acute chest pain, and it could (1) detect unstable angina patients and was confirmed by invasive coronary angiography and low troponin level and (2) detect STEMI patients with persistent ST-segment changes and high troponin level. Moreover, *ABHD4* regulating RNA panel showed consistent differential gene expression in majority of UA patients and STEMI patients. Thus, it shed light on the underlying molecular mechanism associated with the *ABHD4* mRNA panel and its regulatory RNA panel in unstable angina and STEMI patients.

## Figures and Tables

**Figure 1 cells-10-01512-f001:**
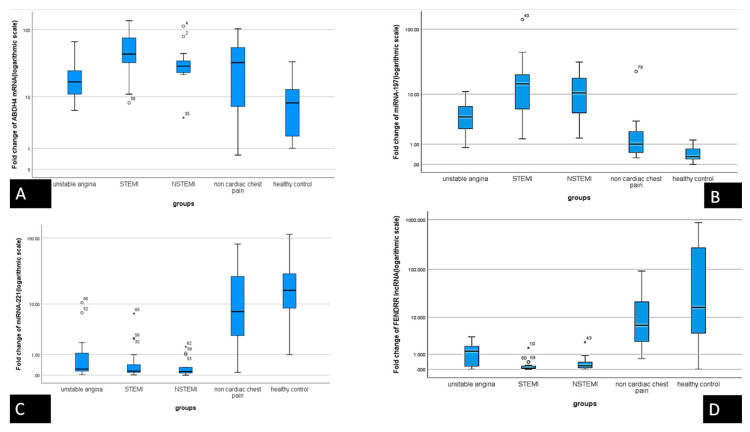
Differential analysis of the results in BOXPLOT represents serum RNA panel expression based on fold change in gene expression at logarithmic scale as measured by qRT-PCR among the groups of the study. (**A**) *ABHD4 mRNA*, (**B**) *MiRNA-197-5p*, (**C**) *miRNA-221-3p*, and (**D**) *FENDRR lncRNA*. The median is represented by line inside the box while the 1st and 3rd quartiles are represented by the top and bottom lines of the box, respectively, and the 5th and 95th percentiles are at the top and bottom whiskers, respectively. *, ° different markers for “out” values (small circle) and “far out” or as SPSS calls them “Extreme values” (marked with a star). SPSS uses a step of 1.5 × IQR (Interquartile range).

**Figure 2 cells-10-01512-f002:**
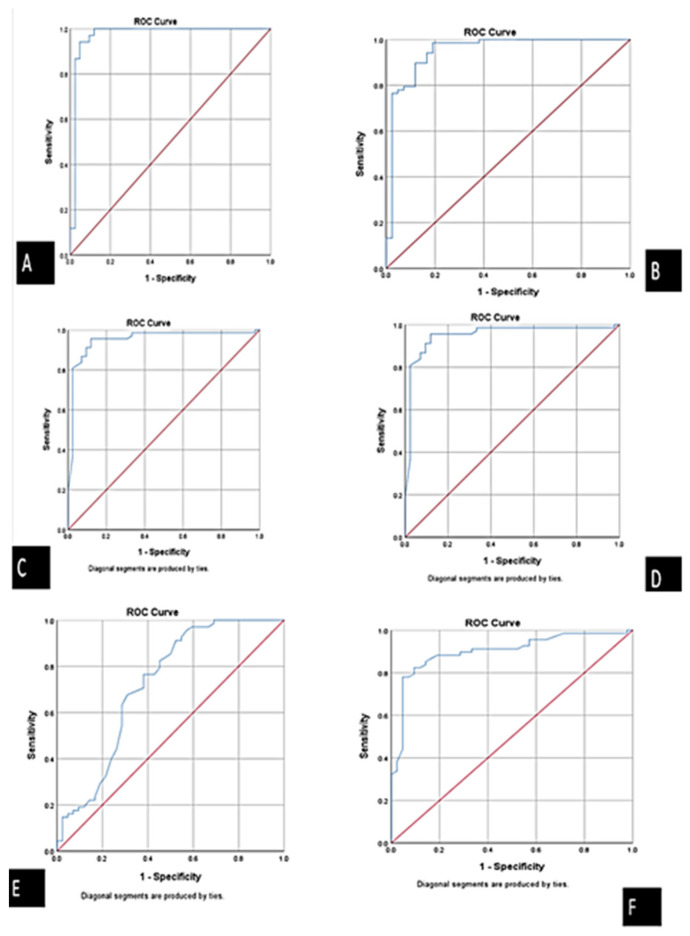
Receiver operator characteristics (ROC) curve presents the diagnostic accuracy of the ABHD4 RNA panel discriminating between ACS and control. (**A**) *ABHD4 mRNA*, (**B**) *MiRNA-197-5p*, (**C**) *miRNA-221-3p*, and (**D**) *FENDRR lncRNA*, (**E**) CK-MB, and (**F**) cardiac troponin.

**Figure 3 cells-10-01512-f003:**
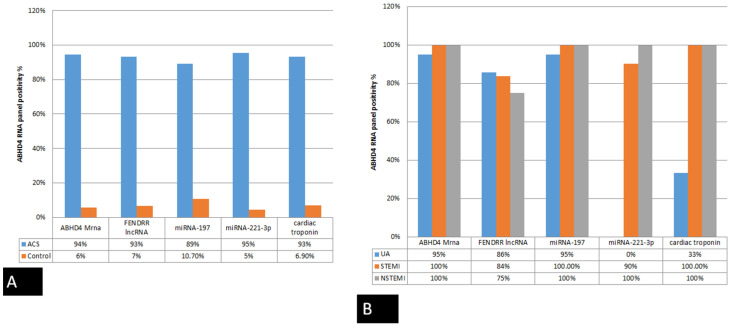
Bar chart shows the positivity rate of the studied parameter: (**A**) between ACS patients and control (non-cardiac and healthy controls), and (**B**) the diagnostic value of the studied parameters in discriminating UA from STEMI and NSTEMI.

**Figure 4 cells-10-01512-f004:**
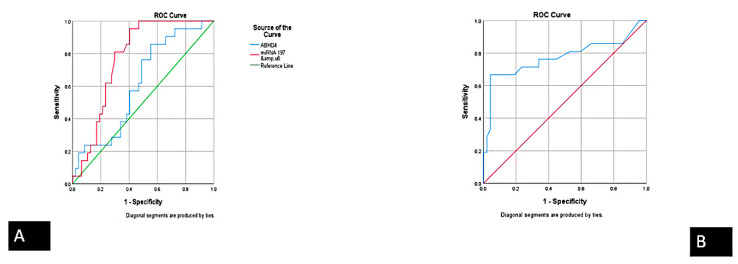
Receiver operator characteristics (ROC) curve presents the diagnostic accuracy of the *ABHD4* RNA panel discriminating between UA and STEMI: (**A**) *ABHD4* mRNA and miRNA-197-5p a (**B**) *FENDRR* lncRNA.

**Figure 5 cells-10-01512-f005:**
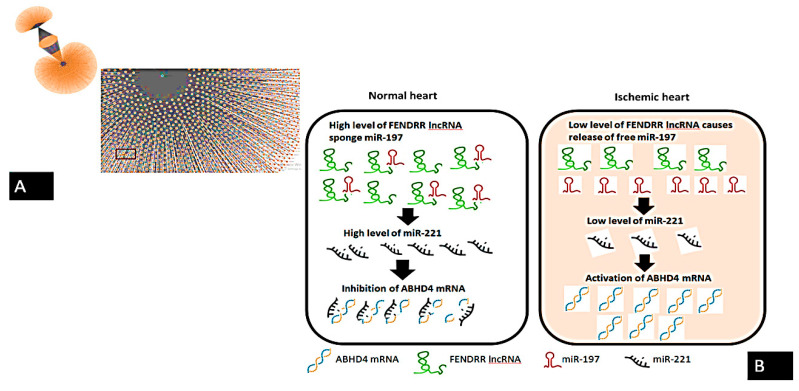
Summary of the molecular signaling of ABHD4-regulating RNA panel in ACS based on putative interactions: (**A**) *ABHD4* interacting with both *hsa-miRNA-221* and *hsa-miRNA-197* as retrieved from miRWalk database, (**B**) Mechanistic signaling of *ABHD4*-regulating RNA panel in ACS. Based on the interaction between *ABHD4* and retrieved RNAs network, we hypothesized that down-regulation of *FENDRR* lncRNA in ischemic heart results in the release of free miR-197 with subsequent downregulation of *miR-221* accompanied with the activation of *ABHD4.*

**Table 1 cells-10-01512-t001:** Differential expression of different parameters among the study groups, in addition to patients with unstable angina versus STEMI.

		i	Median	F	*p*
miRNA-221-3p	unstable angina	21	0.2300	^a^ 13.053	^a^ 0.000
	STEMI	31	0.1500		^b^ 0.00
	NSTEMI	16	0.1300		^c^ 0.874
	non cardiac chest pain	21	7.5000		
	healthy control	21	16.4000		
miRNA 197-5p	unstable angina	21	4.0395	^a^ 7.473	^a^ 0.000
	STEMI	31	14.6000		^b^ 0.001
	NSTEMI	16	10.4500		^c^ 0.001
	non cardiac chest pain	21	1.0000		
	healthy control	21	0.3000		
ABHD4 mRNA	unstable angina	21	8.80000	^a^ 9.142	^a^ 0.000
	STEMI	31	7.80000		^b^ 0.01
	NSTEMI	16	21.40000		^c^ 0.103
	non cardiac chest pain	21	0.04000		
	healthy control	21	0.01000		
FENDRR lncRNA	unstable angina	21	1.30000	^a^ 8.994	^a^ 0.000
	STEMI	31	0.06000		^b^ 0.00
	NSTEMI	16	0.17000		^c^ 0.968
	non cardiac chest pain	21	6.65000		
	healthy control	21	16.50000		
Creatine kinase-MB (CK-MB)	unstable angina	21	17.00	^a^ 10.947	^a^ 0.000
	STEMI	31	44.00		^b^ 0.128
	NSTEMI	16	29.50		^c^ 0.1
	non cardiac chest pain	21	33.00		
	healthy control	21	8.00		
Cardiac Troponin	unstable angina	21	17.00	^a^ 79.243	^a^ 0.000
	STEMI	31	44.00		^b^ 0.154
	NSTEMI	16	29.50		^c^ 0.1
	non cardiac chest pain	21	33.00		
	healthy control	21	8.00		

^a^ Statistics among all groups, ^b^ Statistics UA versus healthy control, ^c^ UNA versus STEMI, *p*-value > 0.05 is considered statistically non-significant, and *p*-value < 0.05 is considered statistically significant. F: One Way Anova. STEMI: ST segment elevation in myocardial infarction.

**Table 2 cells-10-01512-t002:** Performance characteristics of serum laboratory biomarkers.

Biomarker	Sensitivity	Specificity	PPV (Positive Predictive Value)	NPV (Negative Predictive Value)	Accuracy
ABHD4 mRNA	94.4%	97.4%	98.5%	90.5%	95.4%
FENDRR lncRNA	93.3%	90.5%	82.4%	76%	93.3%
miRNA-221-3p	95.4%	86.7%	91.2%	92.9%	91.8%
miRNA-197-5p	89.3%	97.1%	98.5%	81%	91.8%
Cardiac troponin	93.1%	73.1%	79.4%	90.5%	83.6%

**Table 3 cells-10-01512-t003:** Correlation between *ABHD4* RNA panel and cardiac troponin among the study groups.

			MiRNA-221	MiRNA-197	ABHD4	FENDRR	Troponin
Spearman’s rho	*miRNA-221-3p*	Correlation Coefficient	1.000	–0.614 **	–0.596 **	0.678 **	–0.586 **
		Sig. (2-tailed)		0.000	0.000	0.000	0.000
		N	110	110	110	110	110
	*miRNA 197-5p*	Correlation Coefficient	–0.614 **	1.000	0.589 **	–0.586 **	0.641 **
		Sig. (2-tailed)	0.000		0.000	0.000	0.000
		N	110	110	110	110	110
	*ABHD4 mRNA*	Correlation Coefficient	–0.596 **	0.589 **	1.000	–0.587 **	0.596 **
		Sig. (2-tailed)	0.000	0.000		0.000	0.000
		N	110	110	110	110	110
	*FENDRR lncRNA*	Correlation Coefficient	0.678 **	–0.586 **	–0.587 **	1.000	–0.643 **
		Sig. (2-tailed)	0.000	0.000	0.000		0.000
		N	110	110	110	110	110
	Cardiac Troponin	Correlation Coefficient	–0.586 **	0.641 **	0.596 **	–0.643 **	1.000
		Sig. (2-tailed)	0.000	0.000	0.000	0.000	
		N	110	110	110	110	110

** Correlation is significant at the 0.01 level (2-tailed).

## Data Availability

The data reported in this study are available on request from the corresponding authors.

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
