# Peer review of "ABHD4-Regulating RNA Panel: Novel Biomarkers in Acute Coronary Syndrome Diagnosis"

_cells, 2021, doi:10.3390/cells10061512_

Round 1

Reviewer 1 Report

The authors aim to identify novel biomarkers for Acute coronary syndrome (ACS). They investigate ABHD4 mRNA and its associated non-coding RNAs (microRNAs and lncRNAs) by measuring their levels in RNA isolated from serum samples from different patient and healthy cohorts. A recent review (PMID: 32337239) highlighted several ncRNAs associated with ACS. MiR-221 and miR-197 have already been reported to be associated with ACS in previous studies (PMID: 26720041, PMID: 31817254). The lncRNA FENDRR is reported to be crucial for heart development (PMID: 23369715) but so far has not been reported to be associated with ACS specifically. There is a need to identify more reliable biomarkers for ACS which validates the idea of this research work. However, the strategy applied and the tools and methods used to establish ABHD4 and lncRNA Fendrr with ACS remain a major concern. The sample size of the cohort is limited which the authors have already acknowledged. With application of better in silico tools and scientifically sound methods this study has good potential.

Major points:

  1. It is not clear from the text why the authors chose to investigate ABHD4 in the context of ACS. Can they provide a filtering strategy to explain the reason behind selecting ABHD4? A more detailed introduction to the ABHD family of proteins and its relevance for ACS would be helpful for substantiating the scientific idea of this manuscript.

  1. The term “ABHD4-dependent RNA panel” seems misleading. The authors describe that they used in silico prediction tools to identify ABHD4 mRNA and its epigenetic regulators for this study. Hence, to my understanding these are actually ABHD4-regulating RNA panel. 
  1. Similarly, the strategy to select lncRNA Fendrr for this biomarker analysis is again unclear. Identifying binding using multiple sequence alignment is not acceptable. Reliable and established tools should be used to predict lncRNA binding which can be found at https://www.labome.com/method/LncRNA-Research-Resources.html

  1. However, lncRNA Fendrr has not been shown to interact with any of the mentioned miRNAs from this manuscript (PMID: 33513839)

  1. The authors have merely exported screen shots from the GeneCard webpage and added it to the supplementary files. As evident from the screenshots ABHD4 is not specific to heart. It would be more convincing if the authors apply better tools for Gene enrichment analysis. For eg EnnrichR-(https://maayanlab.cloud/Enrichr/) which is an open source tool available online.

  1. In general, the authors could also improve the quality of the figures by cropping away unnecessary parts from the figure panels which include screen shots of the multiple sequence alignment or the GeneCard webpage.

  1. Fig S9 clearly shows that ABHD4 and miR-221-3p binding prediction appears on the 12th page in the entire list of targets predicted. Again raising concerns about why and how these targets were chosed for this study. Can the authors kindly provide robust evidence or a strong reason? Moreover, Luciferase assays should be performed to confirm binding of miRNAs to the 3’ UTR of the ABHD4 mRNA without which the conclusions remain far-fetched.

  1. References 18 and 19; 30 and 31 are the same. Pls correct them and adjust the reference numbers in the text accordingly

  1. Was the serum sample treated with heparinase to avoid interference of heparin during qPCR based RNA quantification ?

  1. Was a spike in used at the time of RNA isolation from the serum samples? This is crucial to normalize the data as difference in RNA quality of the different serum samples also affects the results and interpretation from qPCRs.

  1. What do the authors mean that Ct vales of higher than 36 were considered as negative values? Do they mean that Ct values below 36 were always analyzed. How was the expression of the normalizing gene ACTB1 for the samples with Ct values above 30?

  1. The Table 1 is redundant.

  1. What is meant by “miR-197 & amp; U6” in fig 1 , fig 3 and table 3. Kindly label the axis and tables with appropriate legends. Several places miRNA is misspelled as mRNA in the figure legends, for eg legend of Fig 3 A . Please correct.

  1. Based on supplementary Table 1 , it is clear that the ratio of male to female was quite high. Perhaps the authors can comment if this can influence the interpretation of results in any way for the different cohorts used in this study.

Minor points:

1. Abbreviate UA in the abstract where it is mentioned for the first time. Also unify the labeling as ’UA’ throughout the text and tables.

2. Fig 2- ROC plots should be made of similar sizes to make the figure panel more appealing. Fig 2E and 2F have been copy pasted and not been edited well enough. Kindly pay more attention to the data presentation style.

3. Line 90- The month is missing before 2017

4. Lines 78, 89, 132- there is an extra space

5. Line 177 and 179- sometime a space is given before AUC and sometimes not. Please unify.

6. Line 101- Abbreviate PCI

7. Line 133- taqman should be written with T and M

8. Line 171, 187- mrna written instead of miRNA.

9. Also please be consistent with writing microRNA as “miRNA” or “miRNA-“ or “miR-“ throughout the text and figures and tables.

10. Line 242- should be ‘unstoppable’

11. Reference number 2 is very old. Please replace with a more recent version of ESC guidelines

Author Response

Reviewer 1

Reviewer 1

Comments and Suggestions for Authors

The authors aim to identify novel biomarkers for Acute coronary syndrome (ACS). They investigate ABHD4 mRNA and its associated non-coding RNAs (microRNAs and lncRNAs) by measuring their levels in RNA isolated from serum samples from different patient and healthy cohorts. A recent review (PMID: 32337239) highlighted several ncRNAs associated with ACS. MiR-221 and miR-197 have already been reported to be associated with ACS in previous studies (PMID: 26720041, PMID: 31817254). The lncRNA FENDRR is reported to be crucial for heart development (PMID: 23369715) but so far has not been reported to be associated with ACS specifically. There is a need to identify more reliable biomarkers for ACS which validates the idea of this research work. However, the strategy applied and the tools and methods used to establish ABHD4 and lncRNA Fendrr with ACS remain a major concern. The sample size of the cohort is limited which the authors have already acknowledged. With application of better in silico tools and scientifically sound methods this study has good potential.

Major points:

  1. It is not clear from the text why the authors chose to investigate ABHD4 in the context of ACS. Can they provide a filtering strategy to explain the reason behind selecting ABHD4? A more detailed introduction to the ABHD family of proteins and its relevance for ACS would be helpful for substantiating the scientific idea of this manuscript.
  • As suggested by reviewer, filtering strategy to explain the reason behind selecting ABHD4 has been clarified at the end of introduction. Moreover, detailed introduction to the ABHD family of proteins and its relevance for ACS has been added to the introduction.
  • We have selected ABHD4 mRNA as a crucial player in glycerophospholipid metabolism gene highly correlated to cardiovascular complications especially ACS based on two approaches. Firstly, bioinformatics analysis was used to confirm the expression of ABHD4 mRNA in ACS based on; a)novelty, gene ontology enrichment that confirm its correlation with lipid metabolism regulation, and the basal expression in heart. Taken together, The used public microarray gene expression databases confirmed such selection. The second approach was a literature review [1-4] of the limited data available about the role of ABHD family as predictor of glycerophospholipid in human which has strong correlation with ACS. Both bioinformatics analysis and literature review suggested a possible role of ABHD4 mRNA in ACS.
  • Afterwards, then we retrieved data about miRNA regulation of ABHD4 mRNA and identified (miR-197-5p and miR-221-3p) based on both in silico data and literature reviews[34-39] that affirm that the chosen miRNAs linked to lipid metabolism regulation, targeting ABHD4 mRNA and previously confirmed in ACS.
  • Lastly, we aim to construct Mrna-miRNA-lncRNA integrated genetically linked RNA panel to increase the chance of usefulness of the chosen panel in ACS management and novel implications for targeting ABHD enzymes in the treatment or prevention of lipid metabolic related disease especially ACS. We have selected FENDRR lncRNA based on both in silico data that confirm complementarity binding between the selected RNA panel members and literature search that highlighted its role in inflammation, fibrosis and cardiovascular diseases[40,41].  

  • Dysregulation of lipid metabolism leads to several chronic diseases especially cardiovascular disease. Shedding the light on the enzymatic mechanisms controlling lipid  metabolism is crucial for successful innovative diagnostics and drug discovery in human diseases. α/β hydrolase fold domain  containing (ABHD) proteins family have common roles in lipid metabolism[i]. They includes proteases, esterases, lipases, dehalogenases, and epoxide hydrolases[ii]. ABHD protein family members have the ability to metabolize different glycerophospholipid types that act as key intermediates in cellular signaling and neurotransmission[iii].Recently, ABHD protein family mutations have been linked inborn errors of lipid metabolism with subsequent complications. ABHD protein family is considered as a predictor of plasma phospholipid levels in humans[iv]

  1. The term “ABHD4-dependent RNA panel” seems misleading. The authors describe that they used in silico prediction tools to identify ABHD4 mRNA and its epigenetic regulators for this study. Hence, to my understanding these are actually ABHD4-regulating RNA panel. 

  • As suggested by reviewer, ABHD4-dependent RNA panel has been changed to ABHD4-regulating RNA panel all through MS.

  1. Similarly, the strategy to select lncRNA Fendrr for this biomarker analysis is again unclear. Identifying binding using multiple sequence alignment is not acceptable. Reliable and established tools should be used to predict lncRNA binding which can be found at https://www.labome.com/method/LncRNA-Research-Resources.html
  • We have selected FENDRR lncRNA based on both in silico data that confirm complementarity binding between the selected RNA panel members, GO that affirm its relation to cardiac tissue disorders and literature search that highlighted its role in inflammation, fibrosis and cardiovascular diseases[40,41].  
  • In agreement with the reviewer, we have used the following tools in the highly powerful labome
  • LNCipedia database to check FENDRR lncRNA size(960 bp) and variants(supplementary figure 14)
  • Lnc2atlas database to confirm the chosen FENDRR lncRNA variant  size(960 bp) (supplementary figure 15)
  • Non code database to retrieve FENDRR lncRNA sequence that is used for further alignment.
  • LncTar: an efficient tool for predicting RNA targets of lncRNAs (http://www.cuilab.cn/lnctar)

supplementary figure 19

Supplementary figure 2

  • In agreement with reviewer, we have used Enricher tool to prove the relation between FENDRR and ACS (supplementary figure 13)
  • Moreover , multiple sequence alignment by Claustal program is highly efficient tool for prediction of interaction and for biomarker identification that has been cited several times in many literatures
  • Simossis V, Kleinjung J, Heringa J. An overview of multiple sequence alignment. Curr Protoc Bioinformatics. 2003 Nov;Chapter 3:Unit 3.7. doi: 10.1002/0471250953.bi0307s03. PMID: 18428699.
  • Chenna R, Sugawara H, Koike T, Lopez R, Gibson TJ, Higgins DG, Thompson JD. Multiple sequence alignment with the Clustal series of programs. Nucleic Acids Res. 2003 Jul 1;31(13):3497-500. Doi: 10.1093/nar/gkg500. PMID: 12824352; PMCID: PMC168907.
  • Sievers F, Higgins DG. Clustal Omega, accurate alignment of very large numbers of sequences. Methods Mol Biol. 2014;1079:105-16. Doi: 10.1007/978-1-62703-646-7_6. PMID: 24170397.
  1. However, lncRNA Fendrr has not been shown to interact with any of the mentioned miRNAs from this manuscript (PMID: 33513839)
  • Thanks very much for your valuable comment, the mentioned paper discusses only the miRNA that have been published previously with direct relation with FENDRR lncRNA. Herein, we tried to search for novel predicted genetically linked RNA panel. Thus, both selected miRNAs were not mentioned in previous literature as a direct target of FENDRR-lncRNA.
  1. The authors have merely exported screen shots from the GeneCard webpage and added it to the supplementary files. As evident from the screenshots ABHD4 is not specific to heart. It would be more convincing if the authors apply better tools for Gene enrichment analysis. For eg EnnrichR-(https://maayanlab.cloud/Enrichr/) which is an open source tool available online.
  • Thanks very much for your valuable comment, as suggested by reviewer , we have added ABHD4 mRNA gene enrichment analysis and expression using

EnnrichR tool (supplementary figure 4 )

  1. In general, the authors could also improve the quality of the figures by cropping away unnecessary parts from the figure panels which include screen shots of the multiple sequence alignment or the GeneCard webpage.
  • As suggested by the reviewer, the quality of the figures have been increased by cropping.
  1. Fig S9 clearly shows that ABHD4 and miR-221-3p binding prediction appears on the 12th page in the entire list of targets predicted. Again raising concerns about why and how these targets were chosed for this study. Can the authors kindly provide robust evidence or a strong reason? Moreover, Luciferase assays should be performed to confirm binding of miRNAs to the 3’ UTR of the ABHD4 mRNA without which the conclusions remain far-fetched.
  • As suggested by the reviewer, why and how these miRNAs were chosen for this study has been clarified in the filtration strategy as follows;
  • Afterwards, then we retrieved data about miRNA regulation of ABHD4 mRNA and identified (miR-197-5p and miR-221-3p) based on both in silico data and literature reviews[34-39] that affirm that the chosen miRNAs linked to lipid metabolism regulation, targeting ABHD4 mRNA and previously confirmed in ACS.
  • Moreover, we have tried to choose miRNAs with already established correlation to ACS in previous studies to be previously validated part of ABHD4 regulating RNA panel. Other miRNAs targeting ABHD4 mRNA were either not previously studied in ACS or with no direct relation to lipid metabolism regulation or with no observed expression in cardiac tissues.
  • In agreement with the reviewer, it would be more valuable to take this aspect in our consideration. Luciferase assays seems to be potential experimental technique which we plan to use in future in vitro and in vivo functional study to approve binding of miRNAs to the 3’ UTR of the ABHD4 mRNA. The present work is preliminary step to validate our hypothesis to investigate ABHD4regulating RNA pane differential expression between ACS and control group.
  1. References 18 and 19; 30 and 31 are the same. Pls correct them and adjust the reference numbers in the text accordingly
  • As suggested by the reviewer, Reference 20, and 31 have been deleted and t the reference numbers in the text were adjusted accordingly

  1. Was the serum sample treated with heparinase to avoid interference of heparin during qPCR based RNA quantification ?
  • We have used sera samples not plasma. Thus, there was no need for heparinase

  1. Was a spike in used at the time of RNA isolation from the serum samples? This is crucial to normalize the data as difference in RNA quality of the different serum samples also affects the results and interpretation from qPCRs.
  • Yes we have used, Spike-in control cel-miR-39 (Qiagen, CA, USA), originating from Caenorhabditis elegans, was used for the normalization of miRNAs. It has been added to methodology part
  1. What do the authors mean that Ct vales of higher than 36 were considered as negative values? Do they mean that Ct values below 36 were always analyzed. How was the expression of the normalizing gene ACTB1 for the samples with Ct values above 30?
  • Yes, any samples with Ct more than 36 were considered negative, as regards ACTB1 showed Ct values that spanned from 20 to 30 cycles in all samples
  1. The Table 1 is redundant.
  • As suggested by the reviewer, Table 1 has been modified to be less redundent

  1. What is meant by “miR-197 & amp; U6” in fig 1 , fig 3 and table 3. Kindly label the axis and tables with appropriate legends. Several places miRNA is misspelled as mRNA in the figure legends, for eg legend of Fig 3 A . Please correct.
  • As suggested by the reviewer, mistyping error , & amp; U6” has been deleted. Also, the axis of the tables have been labeled, miRNA misspelling have been corrected in figure legend.
  1. Based on supplementary Table 1 , it is clear that the ratio of male to female was quite high. Perhaps the authors can comment if this can influence the interpretation of results in any way for the different cohorts used in this study.
  • Although some studies declared that symptoms are more common and have greater predictive value in women than in men with myocardial infarction whether or not they are diagnosed using sex‐specific criteria. But on the other hand, according to similar studies ACS was more common in males which goes in hand with our results. moreover, No significant differences between ACS and other 2 control groups ,indicating that control and ACS are age and sex matched that did not affect the results
  • Altaf A, Shah H, Salahuddin M. Gender based differences in clinical and Angiographic characteristics and outcomes of Acute Coronary Syndrome (ACS) in Asian population. Pak J Med Sci. 2019;35(5):1349-1354. doi:10.12669/pjms.35.5.743.
  • Arslanian-Engoren C, Patel A, Fang J, Armstrong D, Kline-Rogers E, Duvernoy CS, Eagle KA. Symptoms of men and women presenting with acute coronary syndromes. Am J Cardiol. 2006 Nov 1;98(9):1177-81. doi: 10.1016/j.amjcard.2006.05.049. Epub 2006 Sep 7. PMID: 17056322.

Minor points:

  1. Abbreviate UA in the abstract where it is mentioned for the first time. Also unify the labeling as ’UA’ throughout the text and tables.
  • As suggested by the reviewer, UA in the abstract has been mentioned as unstable angina. ’UA’ throughout the text and tables have been unified
  1. Fig 2- ROC plots should be made of similar sizes to make the figure panel more appealing. Fig 2E and 2F have been copy pasted and not been edited well enough. Kindly pay more attention to the data presentation style.
  • As suggested by the reviewer, Fig 2- ROC plots has been corrected

  1. Line 90- The month is missing before 2017
  • As suggested by the reviewer, The month has been added as follows, from November 2017 till October 2017
  1. Lines 78, 89, 132- there is an extra space
  • As suggested by the reviewer, extra spaces have been deleted
  1. Line 177 and 179- sometime a space is given before AUC and sometimes not. Please unify.
  • As suggested by the reviewer, space is given before AUC has been unified in Line 177 and 179-
  1. Line 101- Abbreviate PCI
  • As suggested by the reviewer, PCI has been mentioned as Percutaneous Coronary Intervention
  1. Line 133- taqman should be written with T and M
  • As suggested by the reviewer, taqman should be written with T and M in line 133
  1. Line 171, 187- mrna written instead of miRNA.
  • As suggested by the reviewer, they have been corrected
  1. Also please be consistent with writing microRNA as “miRNA” or “miRNA-“ or “miR-“ throughout the text and figures and tables.
  • As suggested by the reviewer, miRNA- has been used throughout the text and figures and tables
  1. Line 242- should be ‘unstoppable’
  • As suggested by the reviewer, the word has been corrected
  1. Reference number 2 is very old. Please replace with a more recent version of ESC guidelines
  • As suggested by the reviewer, Reference number 2 is updated
  • Stephan Windecker, ESC Committee for Practice Guidelines: providing knowledge to everyday clinical practice, Cardiovascular Research, Volume 116, Issue 11, 1 September 2020, Pages e146–e148, https://doi.org/10.1093/cvr/cvaa154

[i]Liu J, Wang L, Harvey-White J, Huang BX, Kim HY, Luquet S, Palmiter RD, Krystal G, Rai R, Mahadevan A, Razdan RK, Kunos G. Multiple pathways involved in the biosynthesis of anandamide. Neuropharmacology. 2008 Jan; 54(1):1-7.

[ii] Nardini M, Dijkstra BW. Alpha/beta hydrolase fold enzymes: the family keeps growing. Curr Opin Struct Biol. 1999;9:732–737.

[iii] Lord CC, Thomas G, Brown JM. Mammalian alpha beta hydrolase domain (ABHD) proteins: Lipid metabolizing enzymes at the interface of cell signaling and energy metabolism. Biochim Biophys Acta. 2013;1831(4):792-802. doi:10.1016/j.bbalip.2013.01.002

[iv] emirkan A, van Duijn CM, Ugocsai P, Isaacs A, Pramstaller PP, Liebisch G, Wilson JF, Johansson A, Rudan I, Aulchenko YS, Kirichenko AV, Janssens AC, Jansen RC, Gnewuch C, Domingues FS, Pattaro C, Wild SH, Jonasson I, Polasek O, Zorkoltseva IV, Hofman A, Karssen LC, Struchalin M, Floyd J, Igl W, Biloglav Z, Broer L, Pfeufer A, Pichler I, Campbell S, Zaboli G, Kolcic I, Rivadeneira F, Huffman J, Hastie ND, Uitterlinden A, Franke L, Franklin CS, Vitart V, Nelson CP, Preuss M, Bis JC, O'Donnell CJ, Franceschini N, Witteman JC, Axenovich T, Oostra BA, Meitinger T, Hicks AA, Hayward C, Wright AF, Gyllensten U, Campbell H, Schmitz G. Genome-wide association study identifies novel loci associated with circulating phospho- and sphingolipid concentrations. PLoS Genet. 2012;8:e1002490.

Reviewer 2 Report

It seems very interesting article, as your conclusion “ABHD4 mRNA dependent RNA panel seems to be novel non invasive biomarkers that can early diagnose ACS and stratify patients severity that could improve health outcome” i’m agree to continue to investigate this topic.

I suggest to investigate if in patients with acute coronary syndrome, can be possible investigate by OCT during coronarographty if there is a interaction with ABHD4 mRNA dependent RNA 32.

As you kow Optical coherence tomography (OCT) is a new imaging technique with higher resolution than the other techniques currently available. In OCT, the refraction of infrared rays produced in contact with the microstructures of biological tissues is used to achieve the definition of microscopic images (10-20 microns), thus allowing the main components of the atheromatous plaque to be identified. Numerous clinical studies have demonstrated the feasibility of the OCT technique in obtaining qualitative information for the characterization of coronary atherosclerotic plaque and for the evaluation of intracoronary stenting.

It could be a topic to investigate in my opinion.

Please add in reference this two article

  1. Review Intern Emerg Med

.2020 Oct;15(7):1193-1199. doi: 10.1007/s11739-020-02422-z. Epub 2020 Jul 3.

Novel biomarkers to assess the risk for acute coronary syndrome: beyond troponins

Andrea Piccioni 1, Federico Valletta 2, Christian Zanza 2, Alessandra Esperide 2, Francesco Franceschi 2

PMID: 32621267 DOI: 10.1007/s11739-020-02422-z

https://pubmed.ncbi.nlm.nih.gov/32621267/

  1. Gut Microbiota and Environment in Coronary Artery Disease

Andrea Piccioni 1, Tommaso de Cunzo 1, Federico Valletta 1, Marcello Covino 1, Emanuele Rinninella 2, Pauline Raoul 3, Christian Zanza 1, Maria Cristina Mele 3 4, Francesco Franceschi 1

PMID: 33923612 PMCID: PMC8073779 DOI: 10.3390/ijerph18084242

https://pubmed.ncbi.nlm.nih.gov/33923612/

Author Response

Reviewer 2

Comments and Suggestions for Authors

It seems very interesting article, as your conclusion “ABHD4 mRNA dependent RNA panel seems to be novel non invasive biomarkers that can early diagnose ACS and stratify patients severity that could improve health outcome” i’m agree to continue to investigate this topic.

I suggest to investigate if in patients with acute coronary syndrome, can be possible investigate by OCT during coronarographty if there is a interaction with ABHD4 mRNA dependent RNA 32.

As you kow Optical coherence tomography (OCT) is a new imaging technique with higher resolution than the other techniques currently available. In OCT, the refraction of infrared rays produced in contact with the microstructures of biological tissues is used to achieve the definition of microscopic images (10-20 microns), thus allowing the main components of the atheromatous plaque to be identified. Numerous clinical studies have demonstrated the feasibility of the OCT technique in obtaining qualitative information for the characterization of coronary atherosclerotic plaque and for the evaluation of intracoronary stenting.

It could be a topic to investigate in my opinion.

Please add in reference this two article

  1. Review Intern Emerg Med

.2020 Oct;15(7):1193-1199. doi: 10.1007/s11739-020-02422-z. Epub 2020 Jul 3.

Novel biomarkers to assess the risk for acute coronary syndrome: beyond troponins

Andrea Piccioni 1, Federico Valletta 2, Christian Zanza 2, Alessandra Esperide 2, Francesco Franceschi 2

PMID: 32621267 DOI: 10.1007/s11739-020-02422-z

https://pubmed.ncbi.nlm.nih.gov/32621267/

  1. Gut Microbiota and Environment in Coronary Artery Disease

Andrea Piccioni 1, Tommaso de Cunzo 1, Federico Valletta 1, Marcello Covino 1, Emanuele Rinninella 2, Pauline Raoul 3, Christian Zanza 1, Maria Cristina Mele 3 4, Francesco Franceschi 1

PMID: 33923612 PMCID: PMC8073779 DOI: 10.3390/ijerph18084242

  • Thanks very much for your valuable comments.
  • In agreement with the reviewer, it would be more valuable to take Optical coherence tomography (OCT) in our consideration in our future study. We plan to extend this study in further larger multicentric one to validate our preliminary results. It will be great chance to carry out such valuable suggestion in ACS.
  • As suggested by the reviewer, both references have been added in in the introduction part
  • Piccioni A, Valletta F, Zanza C, Esperide A, Franceschi F. Novel biomarkers to assess the risk for acute coronary syndrome: beyond troponins. Intern Emerg Med. 2020 Oct;15(7):1193-1199. doi: 10.1007/s11739-020-02422-z. Epub 2020 Jul 3. PMID: 32621267.
  • Piccioni A, de Cunzo T, Valletta F, Covino M, Rinninella E, Raoul P, Zanza C, Mele MC, Franceschi F. Gut Microbiota and Environment in Coronary Artery Disease. Int J Environ Res Public Health. 2021 Apr 16;18(8):4242. doi: 10.3390/ijerph18084242. PMID: 33923612; PMCID: PMC8073779

Round 2

Reviewer 1 Report

I thank the authors for updating the manuscript. I have a few remaining concerns, which are as follows:

Fig 1B- the axis is still incorrect. Pls remove "amp & U6". Kindly thoroughly check your manuscript for such mistakes and typos.

MiRNA molecules bind to the 3' UTR region of their target mRNA. The authors in the supplementary figures (10 and 11) have shown the putative interaction with ABHD4 but there are several mismatches. Mismatches are also visible in the sequence alignment between ABHD4 and lncRNA FENDRR (suppl 16-18). This observation should be discussed as a limitation of this study. Also it will be useful to highlight the interaction specifically in the 3' UTR and also the seed sequence. 

Based on the several references and in silico results it can be assumed that there is a possibility for interaction between ABHD4-miRNAs  (221-3p and 197-5p) and lncRNA FENDRR. However, in the absence of Luciferase assay or even higher sophisticated analysis ( such as CHIP or IP) this interaction can be only considered as a prediction in my opinion. I would suggest the authors to rephrase the manuscript stating that this is a putative interaction and might be a novel biomarker. The text and discussion should highlight this fact. Fig5 should mention that these are putative interactions.

In silico tools are excellent for identifying and predicting possible interactions which the authors have done in this study. However, interactions cannot be validated based only on the in silico analysis. Wet lab experiments are necessary to confirm binding and interactions (for all groups and combinations- mRNA and miRNA, miRNA and lncRNA and finally mRNA and lncRNA) .

Author Response

I thank the authors for updating the manuscript. I have a few remaining concerns, which are as follows:

  • Fig 1B- the axis is still incorrect. Pls remove "amp & U6". Kindly thoroughly check your manuscript for such mistakes and typos.
  • As suggested by the reviewer, Figure 1B axis has been corrected
  • MiRNA molecules bind to the 3' UTR region of their target mRNA. The authors in the supplementary figures (10 and 11) have shown the putative interaction with ABHD4 but there are several mismatches. Mismatches are also visible in the sequence alignment between ABHD4 and lncRNA FENDRR (suppl 16-18). This observation should be discussed as a limitation of this study. Also it will be useful to highlight the interaction specifically in the 3' UTR and also the seed sequence. 
  • As suggested by the reviewer, the following paragraph has been added to the discussion
  • At the in silico data analysis, It was found that both miRNAs could interact with the   selected mRNA and lncRNA with score ˃ 0.95 at CDS binding sites (supplementary figures 10 - 11) that  showed putative interaction with ABHD4 but there are several mismatches. Mismatches are also visible in the sequence alignment between ABHD4 and lncRNA FENDRR (supplementary figure 16-18). Thus, further in-vitro and in-vivo studies are still needed to test and confirm our findings .

  • Based on the several references and in silico results it can be assumed that there is a possibilityfor interaction between ABHD4-miRNAs  (221-3p and 197-5p) and lncRNA FENDRR. However, in the absence of Luciferase assay or even higher sophisticated analysis ( such as CHIP or IP) this interaction can be only considered as a prediction in my opinion. I would suggest the authors to rephrase the manuscript stating that this is a putative interaction and might be a novel biomarker. The text and discussion should highlight this fact. Fig5 should mention that these are putative interactions.
  • As suggested by the reviewer, ABHD4 regulating panel is a putative interaction and might be a novel biomarker has been clarified throughout MS
  • Collectively, ABHD4 mRNA regulating RNA panel based on putative interactions seems to be novel non invasive biomarkers that can early diagnose ACS and stratify patients severity that could improve health outcome.(added to abstract_

  • ABHD4-regulating RNAs retrieval  was based on a putative interaction and might be a novel biomarkers in ACS.(added to discussion)
  • In summary, ABHD4 regulating RNA panel based on putative interactions (marker of glycerophospholipd metabolism, cell damage, apoptosis, inflammation) gene expression was assessed in sera samples collected from patients with acute chest pain at presentation to the hospital, could (1) detect unstable angina patients and was confirmed by invasive coronary angiography and low troponin level and (2) detect STEMI patients with persistent ST-segment changes and increased troponin level.(added to conclusion)
  • Figure 5 :Summary of the molecular signally of ABHD4-regulating RNA panel in ACS based on putative interactions.

  • In silico tools are excellent for identifying and predicting possible interactions which the authors have done in this study. However, interactions cannot be validated based only on the in silico analysis. Wet lab experiments are necessary to confirm binding and interactions (for all groups and combinations- mRNA and miRNA, miRNA and lncRNA and finally mRNA and lncRNA) .
  • In agreement with the reviewer, we plan for further in vitro and invivo functional studies. The current study presents validation for efficiency of ABHD4-regulating RNA panel in ACS versus control to be further validated by more future studies
